# Polydextrose reduces the infection of *Klebsiella pneumoniae* in mice by downregulating the expression of TamA

Lin Su,[1,2] Huajie Zhao,[2] Hafiz Muhammad Ishaq,[3] Ningning Liu,[2] Yalan Yang,[2] Duan Li,[2] Liang Liu,[2] Chuansheng Wang,[1] Fan Yang[1,2]

**ABSTRACT**  Polydextrose (PDX), as a prebiotic, is an extensively branched glucose polymer that can promote the growth of beneficial bacteria in the gut. Recent research indicates that PDX regulates intestinal function and supports immune balance, which helps to protect the gut from pathogenic bacteria. However, scarce research has been found that PDX prevents the host infection through the direct effects on the pathogen. In this study, we developed a mouse model infected with *Klebsiella pneumoniae* by pretreating with PDX, assessed the effect of PDX on *K. pneumoniae* acute infection in mice, and explored its potential mechanisms. We developed a mouse model that is infected with *K. pneumoniae* by pretreating with PDX. Colony counting quantified the *K. pneumoniae* bacterial load in the parenchymal organs of mice. A scanning electron microscope was used to investigate the morphological characteristics of *K. pneumoniae*. The expression level of TamA (translocation and assembly module A) was detected by reverse transcription-polymerase chain reaction (RT-PCR) and western blotting. The CRISPR-Cas9 technique was applied to construct the *tamA* mutant strains (Δ*tamA*) and the *tamA* complement strain (*C-ΔtamA*). The biofilm formation capacity was evaluated by the crystal violet assay. The capsule production was quantified by measuring uronic acid content. In the PDX pretreated model, PDX did not alter the growth characteristics and morphological structure of *K. pneumoniae*. However, it significantly reduces the load of *K. pneumoniae* in the lung, liver, spleen, and intestinal tract of mice, which is related to inhibiting the expression of the outer membrane TamA protein by PDX. In an *in vitro* study, the results indicated that deletion of *tamA* significantly inhibited capsule production and biofilm formation of *K. pneumoniae*, weakened interspecific and intraspecific competitive abilities with other members of the *Enterobacteriaceae* family, and reduced the adhesion ability to Caco-2 and murine lung epithelial (MLE) cells. Compared with the wild strain, PDX treatment and the deletion of *tamA* inhibit the expression of adhesion factors (including *FimH*, *FimC*, *FimD*, and *MrkD*) and the capsule synthesis genes (including *galF*, *wzi*, and *manC*) in *K. pneumoniae*. PDX can prevent the infection of *K. pneumoniae* in mice. The potential mechanism may involve downregulating TamA expression and inhibiting adhesion-related molecules. Therefore, PDX can serve as a potential prebiotic to reduce *K. pneumoniae* infections in both humans and animals.

**IMPORTANCE**  Our findings revealed that polydextrose (PDX) could significantly reduce the load of *Klebsiella pneumoniae* in the lung, liver, spleen, and intestinal tract of mice. The potential mechanism is related to inhibiting the expression of the outer membrane TamA protein by PDX. The deletion of *tamA* significantly inhibited the capsule production and biofilm formation of *K. pneumoniae*, weakened the interspecific and intraspecific competitiveness ability with other members of the *Enterobacteriaceae* family, and reduced the adhesion ability to Caco-2 and MLE cells. Our data suggest that PDX may act as a prebiotic to reduce *K. pneumoniae* infections in humans and animals.

Address correspondence to Chuansheng Wang, chuansonwang@126.com, or Fan Yang, yangf77@163.com.

The authors declare no conflict of interest.

**KEYWORDS** polydextrose, *K. pneumoniae*, TamA, biofilm, interspecies and intraspecies competition, adhesion

*K*lebsiella pneumoniae is an opportunistic bacterial pathogen and is often colonized in the digestive tract, urinary tract, and respiratory tract. When the immune system of the host is weakened, translocation may occur in the body. Therefore, the immuno-deficient hosts are prone to a wide spectrum of infections, such as respiratory system infection, urinary system infection, nervous system infection, abdominal infection, and bloodstream infection (1–4). Among them, respiratory tract infection is particularly serious because it causes persistent tissue damage. In addition, urinary tract infections caused by *K. pneumoniae* are also common hospital-acquired infections, accounting for 25%–40% of nosocomial infections (5, 6). Suppurative liver abscess with meningitis and metastatic endophthalmitis caused by *K. pneumoniae* is also an emerging infectious disease worldwide (7). These studies clarify that with the increasing incidence of *K. pneumoniae* infections, hospitals have adopted a wider variety of drugs and higher dosages for treatment. Due to the irrational use of antibiotics, the drug resistance of *K. pneumoniae* has been increasing year by year (8–10). Furthermore, the widespread use of antibiotics also caused a series of side effects and other fatal diseases in humans, such as intestinal inflammation, flora imbalance, and *Clostridium difficile* infection (11, 12). Therefore, there is an urgent need for new drug development or strategies to replace the existing antibiotics that are currently being used for the treatment of *K. pneumoniae* infection.

Prebiotics are non-digestible food ingredients that enhance host health by selectively stimulating beneficial colonic bacteria (13, 14). The most common prebiotics are conjugated linoleic acid, polyunsaturated fatty acids, inulin, human milk oligosaccharides, phenols, oligosaccharides, and soluble dietary fiber (15). However, the most critical factor in preventing infection is controlling pathogen colonization in the host. Increasing studies show that the use of prebiotics in the early stages of host exposure to pathogens can prevent the occurrence and development of diseases (16, 17). Another reported study elaborates that the prebiotic fructooligosaccharide (FOS) and the prebiotic candidate mannose can reduce the adhesion and biofilm formation ability of *C. difficile* (18). In addition, prebiotics can inhibit the growth of pathogens and the expression of their virulence factors, prevent the invasion of pathogens, improve the gastrointestinal barrier, and regulate mucosal and/or systemic immune responses (19). Therefore, it can be used as a potential drug for the prevention and treatment of some pathogenic infections and has garnered significant attention from researchers.

Polydextrose (PDX), a soluble dietary fiber, is a highly branched, randomly bonded glucose polymer with an average degree of polymerization of 12, ranging from 2 to 120 (20). The molecule contains all possible combinations of α- and β-linked 1→2, 1→3, 1→4, and 1→6 glycosidic linkages, though the 1→6 (both α and β) predominates (21, 22). Due to its complex structure, PDX resists hydrolysis by mammalian digestive enzymes in the small intestine, passing intact into the colon, where it undergoes microbial fermentation before being excreted in the feces (23). PDX fermentation could increase the production of short-chain fatty acids (acetate, propionate, and butyrate) in the colon (24). Specifically, PDX-induced butyrate enhances colonic mucosal integrity and significantly contributes to host protection and defense mechanisms (25). In addition, the intake of polydextrose has a certain impact on the composition of intestinal microbiota. Mariane et al. found that the intake of PDX significantly inhibited the abundance of phylum *Firmicutes* and raised the abundance of phylum *Bacteroides* (26). Tina et al. also revealed that the intake of PDX altered the host intestinal pH and inhibited the growth of pH-sensitive gram-negative bacteria (27).

The translocation and assembly module (TAM) is a protein channel composed of TamA and TamB, where TamA acts as a primary functional component (28, 29). The deletion of *tam* can reduce the flagellum formation, movement speed, and invasive ability of *Edwardsiella tarda* (30). Recent studies explain that TAM is closely related to

the assembly of outer membrane proteins, which play a pivotal role in bacterial lifespan activities (29–31). Jung and colleagues have demonstrated that TAM deletion increases carbapenem-resistant *K. pneumoniae* (CR-Kp) outer membrane protein permeability under stress conditions and enhances sensitivity to high-molecular-weight antimicrobials, and stress-induced sensitization enhances clearance of TAM-deficient CR-Kp from the gut lumen following fecal microbiota transplantation and from infection sites following pulmonary or systemic infection (32). Therefore, TAM was selected as a study target for developing novel agents to control *K. pneumoniae* infection in hosts.

As an effective prebiotic, PDX can promote the growth of *Bifidobacterium* and *Lactobacillus* while inhibiting the colonization of harmful bacteria (33–35). However, prior PDX studies mainly focused on enhancing colonization resistance to pathogens by altering the internal environment of the host. Currently, few studies report that PDX affects colonization and infection through its direct effect on pathogens. In the current study, *K. pneumoniae* was co-cultured with PDX. Our findings indicate that PDX modulates the expression of the *K. pneumoniae* membrane protein TAM, consequently altering biological functions and reducing respiratory and intestinal tract infections in mice. The current study presents potential strategies for the development of agents that prevent and mitigate *K. pneumoniae* infections.

## MATERIALS AND METHODS

### Animals

Seven to 8-week-old SPF C57BL/6 male mice were purchased from SKBEX Biotechnology Co., Ltd. (Anyang, China; LISC20200005). Mice were maintained in a standard room with 20°C–22°C temperature and 50% ± 5% relative humidity. Animals were housed in filter-topped cages and were free to drink and eat.

### Grouping and infection of mice

After 1 week of acclimatization, the mice were randomly divided into six groups: control (Con-KPI and Con-KPN), 5%PDX (5%PDX-KPI and 5%PDX-KPN), and ΔtamA (ΔtamA-KPI and ΔtamA-KPN). Con-KPI, 5%PDX-KPI, and ΔtamA-KPI groups were given oral gavage with $1 \times 10^8$ CFU of wild-type (WT) *K. pneumoniae*, $1 \times 10^8$ CFU of *K. pneumoniae* pretreated with 5% PDX, and $1 \times 10^8$ CFU *tamA* mutant *K. pneumoniae,* respectively. Con-KPN, 5%PDX-KPN, and ΔtamA-KPN groups were given intranasal inoculation with $1 \times 10^7$ CFU wild-type *K. pneumoniae*, $1 \times 10^7$ CFU *K. pneumoniae* pretreated with 5% PDX, and $1 \times 10^7$ CFU *tamA* mutant *K. pneumoniae*, respectively. Fecal samples were collected after 24 h and 48 h of infection. The mice were sacrificed after the fecal, lung, liver, spleen, and kidney samples collection.

### Pretreatment of *K. pneumoniae* with PDX and culture

*K. pneumoniae* (K5), a clinical isolate from bacteremia patients, was used in this study, whose gene type belongs to ST11. This strain belongs to a classical *K. pneumoniae*, whose hypermucoviscous phenotype is negative. The mutant strains described in this study were generated based on K5. Unless otherwise stated, all bacteria were grown in Luria-Bertani (LB) broth or on LB agar at 37°C. In the case of antibiotic screening of bacteria, the following antibiotics were added to the media: spectinomycin (100 µg/mL), apramycin (50 µg/mL), and kanamycin (50 µg/mL). A fresh culture of bacteria at late exponential phase ($OD_{600}$ = 0.8–1.0) was added to LB containing PDX, incubated at 37°C for 5 h, centrifuged at 12,000 rpm for 4 min, and washed twice with phosphate-buffered saline (PBS) to obtain bacterial precipitation for subsequent experiments. LB containing PDX is added with 0.05 g/mL PDX based on conventional LB (10 g tryptone, 5 g yeast extract, and 10 g NaCl dissolved in 1 L deionized water).

To quantify bacterial burden in the gastrointestinal tract and respiratory tract, animal fecal and lung tissue samples were collected at 24 h and 48 h after infection with *K.*

*pneumoniae*. These feces were weighed and homogenized with PBS, and the actual number of bacteria was continuously gradient diluted to *Klebsiella* selective agar plates (MacConkey inositol adonitol carbenicillin agar) for standard colony count.

## Construction of the *tamA* mutant and complement strains

To construct the pSGK TamA plasmid required by Δ*tamA*, TamA single guide RNA (sgRNA) was designed by CasDesigner software and sent to the commercial company for synthesis (Shanghai Sangon Biotech, Limt. co). The synthesized sgRNA (TamA space F/TamA space R) was connected with the pSGK plasmid linearized by the Basal HFv2 enzyme to obtain the pSGK TamA plasmid, which was identified using pSGK F/pSGK R. To construct pGEM-Teasy TamA, which expresses recombinant C-Δ*tamA*, *tamA* was amplified by PCR with primers C-TamA-F/C-TamA-R. The pGEM-Teasy TamA plasmid was obtained by ligating PCR products with the T-A cloning vector PGEM-Teasy plasmid. To construct Δ*tamA*, CRISPR-Cas9 technology was used to completely knock out the *tamA*. First, the pCas plasmid was introduced into *K. pneumoniae* and identified by primers pCas F/pCas R. Then pSGK TamA plasmid and single-stranded DNA were simultaneously introduced into *K. pneumoniae* containing the pCas plasmid, and the bacteria were coated on an LB agar plate containing Apr antibiotic after recovery and cultured overnight at 28°C. Using TamA-F/TamA-R and TamA-jd-F/TamA-jd-R for identification, Δ*tamA* was obtained. C-Δ*tamA* was obtained by electro-transfer of the pGEM Teasy TamA plasmid based on Δ*tamA* and was identified by TamA-F/TamA-R. The primers used in this study are shown in Table 1.

## Determination of biofilm formation

*K. pneumoniae* was cultured at 37°C until its $OD_{600} = 0.5$, and 2 μL was added to a 96-well microtitration plate containing 200 μL LB. LB without bacteria in each group was used as a control. After static incubation at 37°C for 48 h, the optical density value ($OD_1$) at 570 nm was determined by an enzyme labeling instrument, and the control hole was $OD_{10}$. Planktonic cells were then removed, and wells were washed thrice with PBS (pH 7.2–7.4) to remove unattached cells. Methanol (200 μL, 99%) was added to each well for 20 min. After discarding the methanol, the plates were dried at room temperature. The attached cells were stained for 30 min with 200 μL of 0.5% crystal violet. Unbound dye was removed with water. Plates were dried upside down in a hot-air oven. Ethanol (200 μL) was added to each well to dissolve the bound dye. The optical density value ($OD_2$) at 570 nm was determined, and the optical density of the control hole was $OD_{20}$. Biofilm formation capacity was calculated by using the following formula: $BI = (OD_2 - OD_{20})/(OD_1 - OD_{10})$.

## Quantification of capsule production

The capsule production was quantified by measuring uronic acid content as previously described (36). Briefly, 500 μL bacterial culture was mixed with 100 μL 1% Zwittergent-100 mM citric acid buffer at 50°C for 20 min. Bacterial cells were pelleted by centrifugation, and uronic acid was extracted by precipitating with ethanol and then dissolved in tetraborate/sulfuric acid. The absorbance at 520 nm of uronic acid was detected by the addition of 3-phenylphenol for a 5 min incubation at room temperature. The uronic acid content was quantified through a standard curve of glucuronic acid and presented as micrograms per $OD_{600}$.

## Interspecific and intraspecific competitive growth assays

Cultures of predator (*K. pneumoniae* $K_5$) and prey strains (*Pseudomonas aeruginosa*, *Salmonella* Typhimurium, *Escherichia coli* 100, *K. pneumoniae* 1 [20 $K_1$, ST764], *K. pneumoniae* 2 [20 $K_2$, ST2237], and *K. pneumoniae* 13 [20 $K_{13}$, ST23]) were mixed at a multiplicity of infection (MOI) of 10:1 ($1 \times 10^5$ CFU:$1 \times 10^4$ CFU), and 20 μL of the mixed bacterial culture was spotted onto LB agar for 5 h. Bacterial spots were harvested, and

**TABLE 1** Primers used in this study

| Primer | Sequence (5′–3′)$^a$ |
|---|---|
| *pCas* F | CTAAATGCCGTCGTTG |
| *pCas* R | CTTATGCTGCTCCACA |
| *TamA* space F | GAATGTGCGTTTGCAGGTTG |
| *TamA* space R | CTTACACGCAAACGTCCAAC |
| *pSGK* F | CGCAGCGAATGTGCGTTTGC |
| *pSGK* R | CCAGAGCGATGAAGTCACGC |
| *TamA*-jd-F | CATCGGGACGGTGCTCAA |
| *TamA*-jd-R | AGCGGTGGCGGTCATAGA |
| *TamA*-F | GGTCGGATTGGGCGTATA |
| *TamA*-R | GGGCGTCGAGCTTCATCT |
| C-*TamA*-F | ccatggcggccgcgggaattcGTGCTAAAAATCCGCCAGACCGA |
| C-*TamA*-R | TcaatcgaatttatggtcgacTCATAATTCAGGCCCCAGACCGA |
| RT 16s-F | TGCCTGATGGAGGGGGATAA |
| RT 16s-R | TTCCAGTGTGGCTGGTCATC |
| RT *TamA*-F | TTATTACGAGCCGACGATTGAA |
| RT *TamA*-R | ATCACCTCGGTTCCGCCAAT |
| RT *FimH*-F | TATGATTCACGGACCGATAAACCC |
| RT *FimH*-R | TGGAAGGAGTCGCTATTGTAGTTGT |
| RT *FimC*-F | GTCATCACTCCTCCGCTGTT |
| RT *FimC*-R | ATTCGCCTTCACGTTCATCC |
| RT *FimD*-F | CCACGGCGAGGTGTCAAT |
| RT *FimD*-R | GGAAGCCATCCGCTTCTT |
| RT *MrkD*-F | ATCCCGTTTCACTCAGGC |
| RT *MrkD*-R | ATGTCCGATGTGGGTTCCAT |
| RT *galF*-F | TGCATA**TCTAGA**CTGTACGACTGCGGTATGTGT |
| RT *galF*-R | CAGTAC**GAATTC**TTTGTGGCCGGCAGCATATGC |
| RT *wzi*-F | TGCATATCTAGACGGTAATTGATAATTCATATT |
| RT *wzi*-R | CAGTACGAATTCTGGGCTCCCAGGGAGGAAAGC |
| RT *manC*-F | TGCATATCTAGAGTGCGCACACCTATAAGCGTA |
| RT *manC*-R | CAGTACGAATTCGCTCGCGAGACATCGGCCAGA |

$^a$Lowercase letters are the homologous arms of primers.

the CFU per milliliter of surviving prey and predator strains were measured by plating serial dilutions on selective agar. The output/input ratio of the prey-to-predator strains was interpreted as survival.

## Cell adhesion assays

Human colorectal epithelial Caco-2 cells and mouse lung epithelial MLE cells were inoculated into 24-well plates in minimum essential medium (MEM) with 10% fetal bovine serum at a seeding density of $2 \times 10^5$ cells/well. One hour before the adhesion experiments, the medium was replaced with prewarmed MEM without penicillin and streptomycin. *K. pneumoniae* at the exponential growth stage was centrifuged and washed thrice with PBS. After the cells adhered to the wall, 1 mL of bacteria solution was added to each well to make the MOI 100:1. It was then incubated at 37°C in 5% $CO_2$ for 5 h, then washed with PBS thrice, and lysed with 1% Triton X-100 to release bacteria. Colony coating was counted after multiple dilutions.

Fluorescence microscopy is used to further observe the adhesion of bacteria to cells. The digested cells were added to a six-well plate ($5 \times 10^5$ cells/well) and cultured at 37°C and 5% $CO_2$ for 24 h. The floating cells were washed off with PBS. Five hundred microliters of Cytotrace TMO range dye solution was added to each well, incubated at 37°C for 20 min, and rinsed with PBS three times. Two milliliters of *K. pneumoniae* solution containing green fluorescent plasmid (Ptd103luxI-sfGFP) was added to each well to make MOI 100:1. The bacteria were cultured at 37°C and 5% $CO_2$ for 5 h. Then, the

bacteria were washed with PBS three times. The adhesion of the bacteria was observed under the fluorescence microscope.

## Quantitative real-time PCR analysis

To detect the expression of *TamA*, *FimH*, *FimC*, *FimD*, *MrkD*, *galF*, *wzi*, and *manC*, total RNA of wild-type *K. pneumoniae*, *K. pneumoniae* co-cultured with PDX, and *tamA* mutant *K. pneumoniae* were extracted. The RT-PCR system was used to detect the expression of different genes. RT-PCR primers are shown in Table 1.

## Western blotting

TamA expression levels of *K. pneumoniae* pretreated with PDX were detected by applying western blotting. *K. pneumoniae* was inoculated into LB containing PDX and incubated at 37°C for 5 h. The bacteria were lysed by Bacterial Protein Preparation Lysate, and the protein concentration was determined by Bradford assay. Twenty micrograms of protein was obtained from each group and separated using 10% SDS-PAGE. The proteins were then transferred to polyvinylidene fluoride membranes and blocked with 5% milk at room temperature for 1 h. Primary antibodies, TamA pAb (Murine polyclonal antibody, produced by our laboratory), and GAPDH mAb (Murine monoclonal antibody, purchased from Servicebio, China) were incubated with the membrane overnight at 4°C. The membrane was then incubated with anti-mouse secondary antibodies (Servicebio, China) for 1 h at room temperature on a shaker. Following three washes with PBS, the protein bands were visualized using enhanced chemiluminescence and the Bio-Imaging LAS 1000 plus.

## Scanning electron microscope

The effect of PDX on the morphology and structure of *K. pneumoniae* was observed by scanning electron microscope. A 0.5 McFarland standard of *K. pneumoniae* pretreated with PDX was transferred into Eppendorf tubes containing 2.5% glutaraldehyde fixative at 4°C, followed by dehydration in graded ethanol. The sample was then air-dried, sputter-coated with gold, and examined under the scanning electron microscope.

## Statistical analyses

All statistical analyses in this study were performed by GraphPad Prism 9.0. A nonparametric *t*-test and one-way analysis of variance (ANOVA) were used to compare the different groups. A *P*-value of 0.05 or less was considered statistically significant. A Tukey *post hoc* test was applied to determine pairwise differences where appropriate.

## RESULTS

### PDX reduces the infection of *K. pneumoniae* in mice

To investigate the effect of PDX on the infection of *K. pneumoniae* in mice, *K. pneumoniae* and PDX were co-cultured for 5 h. The mice were then infected by oral gavage or intranasal inoculation; the experimental design is shown in Fig. 1a. The results showed that pretreatment of *K. pneumoniae* with 5%PDX significantly reduced lung load in mice compared with the control group at both 24 h and 48 h (Fig. 1b). At the same time, the amount of colonization in the liver, spleen, and kidney of mice also decreased significantly in the PDX group vs the control group (Fig. 1c through e). After the oral gavage infection, the intestinal load of *K. pneumoniae* in the 5%PDX-KPN group was significantly lower than that in the control group at 24 and 48 h (Fig. 1f), and the load of *K. pneumoniae* in the liver, spleen, and kidney was also significantly lower than that in the control group (Fig. 1g through I). These data revealed that PDX can reduce *K. pneumoniae* infection in mice.

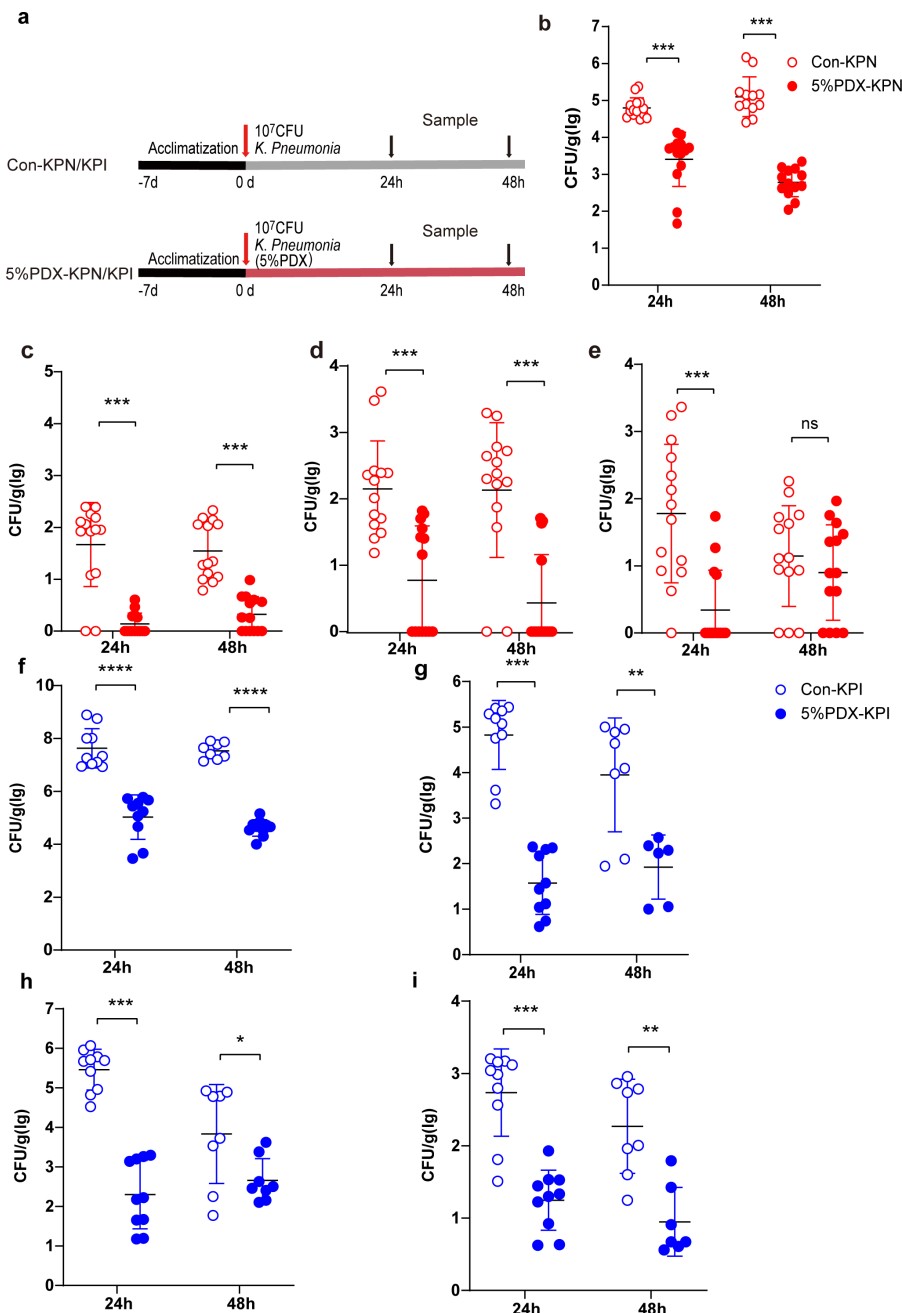

**FIG 1** Effect of PDX on infection of *K. pneumoniae* in mice. (a) The design procedure of the experiment. After infecting mice with WT and WT(P) strains by intranasal inoculation for 24 h and 48 h, respectively, the load of *K. pneumoniae* in the lungs (b), liver (c), spleen (d), and kidney (e) (*n* = 6–8). After infecting mice with WT and WT(P) strains by oral gavage for 24 h and 48 h, respectively, the load of *K. pneumoniae* in the intestine (f), liver (g), spleen (h), and kidney (i) (*n* = 6–8). Data are represented as means ± SEM. Significance was assessed using an unpaired *t*-test. *P < 0.05; **P < 0.01; ***P < 0.001; ****P < 0.0001; ns indicates not significant.

## PDX decreases the expression of TamA of *K. pneumoniae*

To explore the effect of PDX on the biological characteristics of *K. pneumoniae*, we first examined its impact on the growth characteristics of *K. pneumoniae*. The results demonstrate that the growth curve of the WT(P) group (the wild type strains co-cultured with PDX) is consistent with that of the WT group, indicating that PDX did not affect the growth of *K. pneumoniae* (Fig. 2a). Then, we consider whether PDX affects the surface

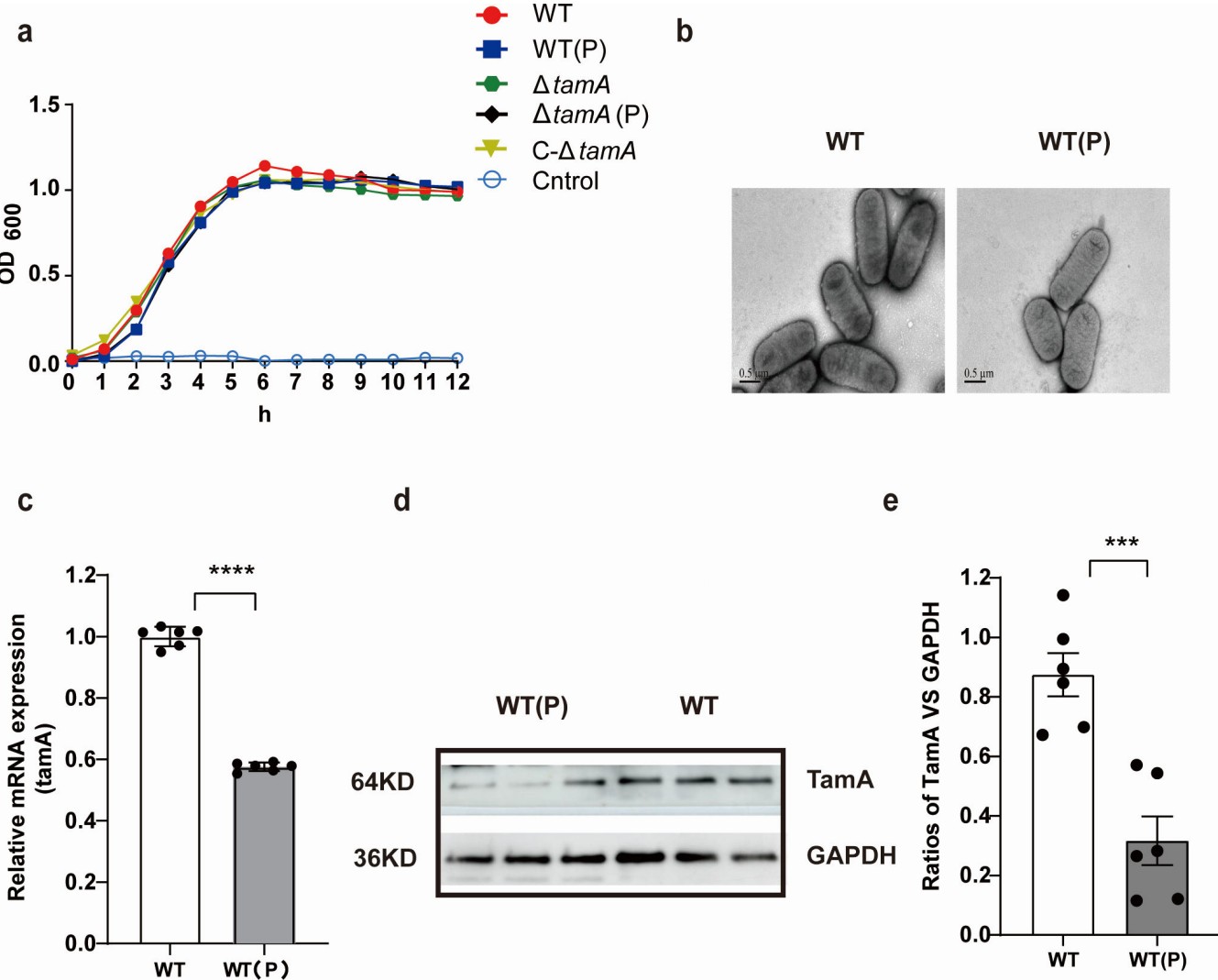

**FIG 2** Effect of PDX on the growth and TamA expression of *K. pneumoniae*. (a) Growth curves of WT strain, WT strain pretreated with PDX [WT(P)], *tamA* mutant strain (Δ*tamA*), *tamA* mutant strain pretreated with PDX [Δ*tamA*(P)], and *tamA* complementary strain (C-Δ*tamA*). (b) Electron microscope photos of WT and WT(P). (c) TamA mRNA expression level of WT and WT(P) strains. (d) The expression levels of TamA protein in the WT and WT(P) strains. (e) Quantitative western blot data are represented as means ± SEM. The discrepancy of growth curves was compared with the Gehan-Breslow Wilcoxon test. The significance of TamA mRNA expression was assessed using an unpaired *t*-test *$P < 0.05$; **$P < 0.01$; ***$P < 0.001$; ****$P < 0.0001$; ns indicates not significant.

structure of *K. pneumoniae*. Scanning electron microscopy shows that PDX does not change the morphological structure (Fig. 2b). TAM is a transporter on the cell membrane, which is involved in the assembly and correct insertion of outer membrane proteins (36). Previous studies have shown that TamA expression is associated with the colonization of *K. pneumoniae* in mice (32). Therefore, we investigate whether PDX affects the expression of this protein. The RT-PCR results indicate that PDX can significantly reduce TamA expression (Fig. 2c). Then, western blotting was further used to verify its expression at the protein level, and the results of western blotting are consistent with those of RT-PCR (Fig. 2d and e). Subsequently, we used CRISPR-Cas9 technology to construct Δ*tamA* and C-Δ*tamA* strains to further confirm the effect of TamA on the growth characteristics of *K. pneumoniae*. The finding shows that the deletion of *tamA* does not affect the growth of *K. pneumoniae*, and the co-culture of PDX and the Δ*tamA* strain also does not affect its growth characteristics (Fig. 2a). Overall, the gathered findings indicate that PDX does not affect the growth of *K. pneumoniae*, yet it significantly reduces the abundance of the membrane protein TamA.

## PDX reduces the biofilm formation and the capsule production of *K. pneumoniae* by downregulating the expression of TamA

Biofilm is a very important factor in bacterial infection, so we tested the effect of PDX on the biofilm formation ability of *K. pneumoniae*. The results demonstrate that the co-culturing of PDX and *K. pneumoniae* can significantly reduce its biofilm-forming ability (Fig. 3a). Then, we compared the biofilm formation of Δ*tamA* and C-Δ*tamA* strains. The results show that when compared with WT, the deletion of *tamA* significantly reduces the biofilm formation ability. After the complementary *tamA* gene, the trend of the biofilm formation was recovered (Fig. 3b). However, there is no significant difference in biofilm formation capacity after co-culture of PDX and Δ*tamA* compared with Δ*tamA* (Fig. 3a). The polysaccharide capsule is the primary virulence factor of *K. pneumoniae*, and the uronic acid content is frequently used as an indicator to quantify capsule production (36). In the present study, we tested the uronic acid concentrations in the co-culture of PDX and *K. pneumoniae* along with *tamA* gene deletion strains. The results show that the uronic acid content of the strains co-cultured with PDX is approximately half that of the WT strains (Fig. 3c). The Δ*tamA* mutants also produce uronic acid at only half of the WT levels, and complementary *tamA* gene does not recover the amount of uronic acid production (Fig. 3c). The PDX-treated Δ*tamA mutants* also display a similar amount of uronic acid with Δ*tamA* mutants, which is significantly lower than that of the WT strain (Fig. 3c). All the results suggest that PDX treatment reduces the capsule production and the biofilm formation of *K. pneumoniae*, likely through downregulating the expression of TamA.

## PDX decreases the interspecies competition of *K. pneumoniae* by downregulating the expression of TamA

Using *K. pneumoniae* as the predator strain, *P. aeruginosa*, *S.* Typhimurium, and *E. coli* 100 as prey strains, interbacterial competition experiments were performed *in vitro*. The results showed that compared with the control group, PDX treatment of *K. pneumoniae* significantly reduced its competitiveness growth with *P. aeruginosa*, *S.* Typhimurium, and *E. coli* 100 (Fig. 4a through c), especially with *S.* Typhimurium (Fig. 4b). To verify whether TamA participated in the competition between *K. pneumoniae* and other *Enterobacteriaceae*, we tested the competitiveness of Δ*tamA* and C-Δ*tamA* strains against *P. aeruginosa*, *S.* Typhimurium, and *E. coli* 100. The results demonstrate that compared with the wild-type strains, the competitiveness of Δ*tamA* strains is significantly reduced against the three *Enterobacteriaceae*. The ability of the *C-ΔtamA* strain to kill the three *Enterobacteriaceae* was partly restored by complementation (Fig. 4d through F). There is no significant difference in competitiveness between the PDX-treated Δ*tamA* strain and the Δ*tamA* strain against the three *Enterobacteriaceae* (Fig. 4a through c). All these results indicate that PDX can reduce competition between *K. pneumoniae* and other *Enterobacteriaceae* by downregulating the expression of TamA.

## PDX weakens the intraspecies competition of *K. pneumoniae* by reducing the expression of TamA

Using *K. pneumoniae* (K5) as the predator strain, *K. pneumoniae* 20-K1, 20-K2, and 20-K13 strains (these strains contain an apramycin-resistant plasmid) were employed as prey to detect intraspecies competition ability. The results show that the wild-type strain has a stronger ability to eliminate the 20-K1, 20-K2, and 20-K13 strains compared with the PDX-treated *K. pneumoniae* (Fig. 5a through c). To determine whether TamA participated in the intraspecies competition of *K. pneumoniae*, we evaluated the competitiveness of Δ*tamA* and C-Δ*tamA* strains with 20-K1, 20-K2, and 20-K13. The results elucidate that the growth of 20-K1, 20-K2, and 20-K13 is significantly inhibited when co-cultured with the wild-type strain compared with the Δ*tamA* strain. Moreover, the ability of the C-Δ*tamA* strain to kill 20-K1, 20-K2, and 20-K13 shows a recovery trend after *tamA* gene complementation (Fig. 5d through f). There was no significant difference in the competitiveness

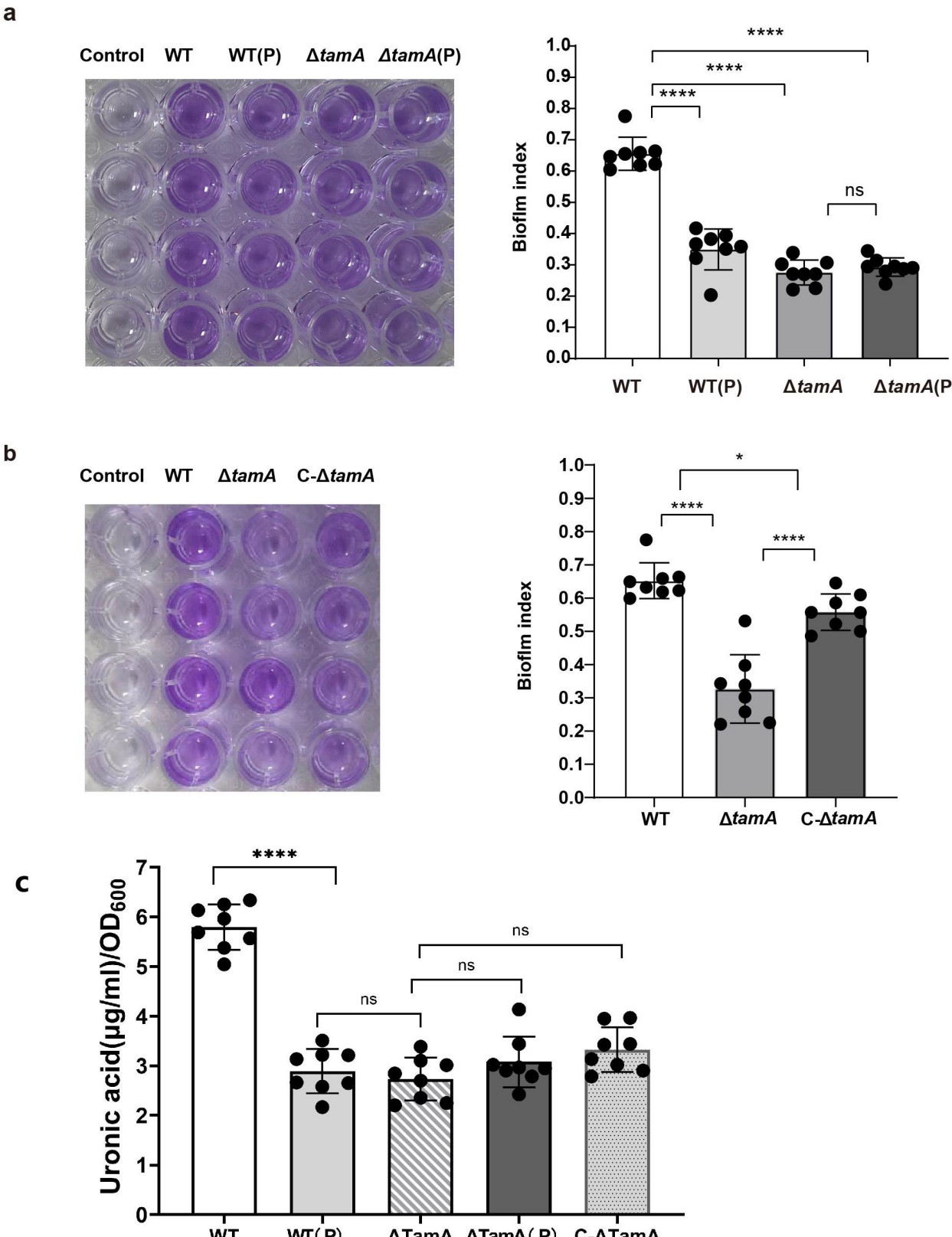

**FIG 3** PDX reduces the biofilm formation and the capsule production of *K. pneumoniae* by inhibiting the expression of TamA. (a) Biofilm formation of WT, WT (P), Δ*tamA*, and Δ*tamA* (P) strains. (b) Biofilm formation of WT, Δ*tamA*, and C-Δ*tamA* strains. (c) The capsule production of WT, WT (P), Δ*tamA*, Δ*tamA* (P), and C-Δ*tamA* strains. Data are represented as means ± SEM. Significance was assessed using a one-way ANOVA test followed by Tukey's *post hoc* test for multiple comparisons. *$P < 0.05$; **$P < 0.01$; ***$P < 0.001$; ****$P < 0.0001$; ns indicates not significant.

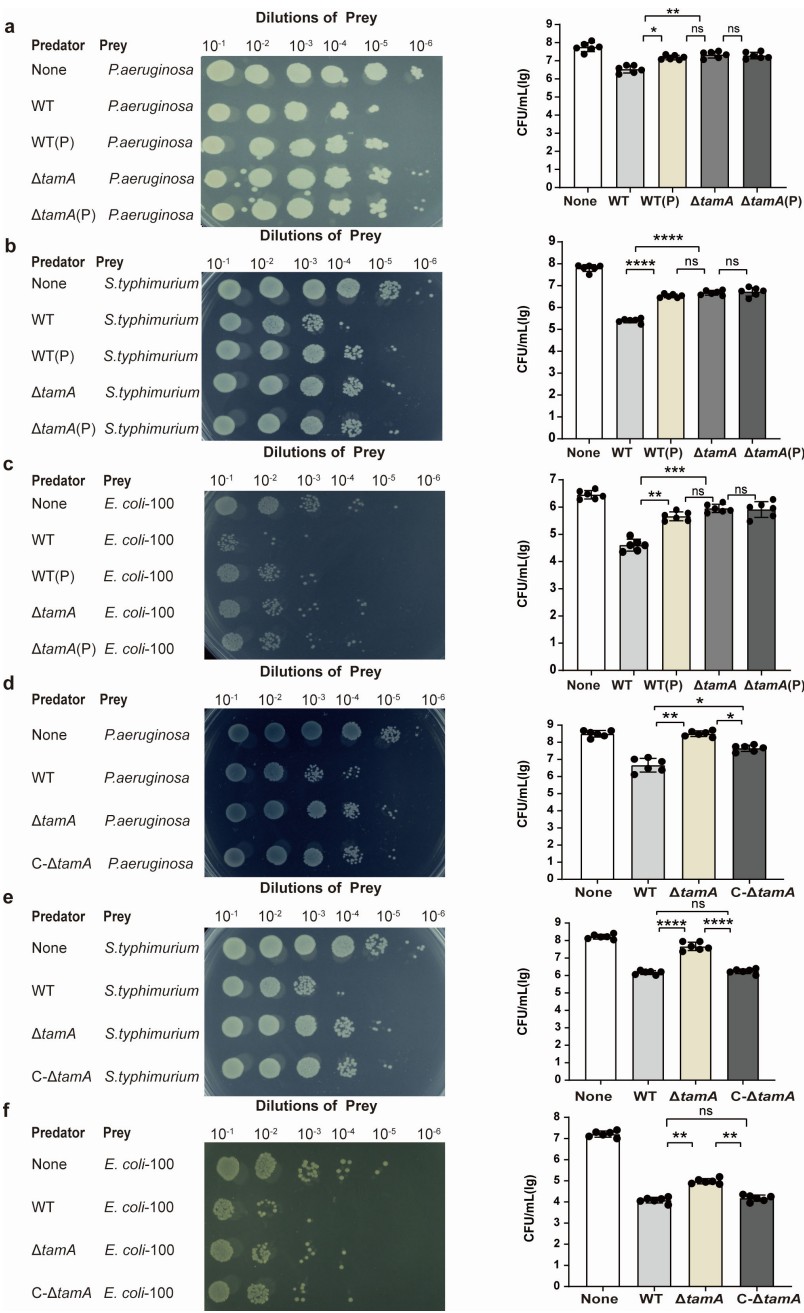

**FIG 4** PDX affects the interspecies competition of *K. pneumoniae* by reducing the expression of TamA. *P. aeruginosa* (a), *S.* Typhimurium (b), and *E. coli*100 (c) were incubated with WT, WT (P), Δ*tamA*, and Δ*tamA* (P) strains for 5 h to determine the survival rates of the prey strain. The survival rate of the prey strain was tested after *P. aeruginosa* (d), *S.* Typhimurium (e), and *E. coli* 100 (f) were incubated with WT, Δ*tamA*, and C-Δ*tamA* strains for 5 h. Data are represented as means ± SD. Significance was assessed using a one-way ANOVA test followed by Tukey's *post hoc* test for multiple comparisons. *$P < 0.05$; **$P < 0.01$; ***$P < 0.001$; ****$P < 0.0001$; ns indicates not significant.

of Δ*tamA* (P) and Δ*tamA* with three kinds of *K. pneumoniae* (Fig. 5a through c). In summary, these results revealed that PDX can weaken the intraspecies competition ability of *K. pneumoniae* by reducing the expression of TamA.

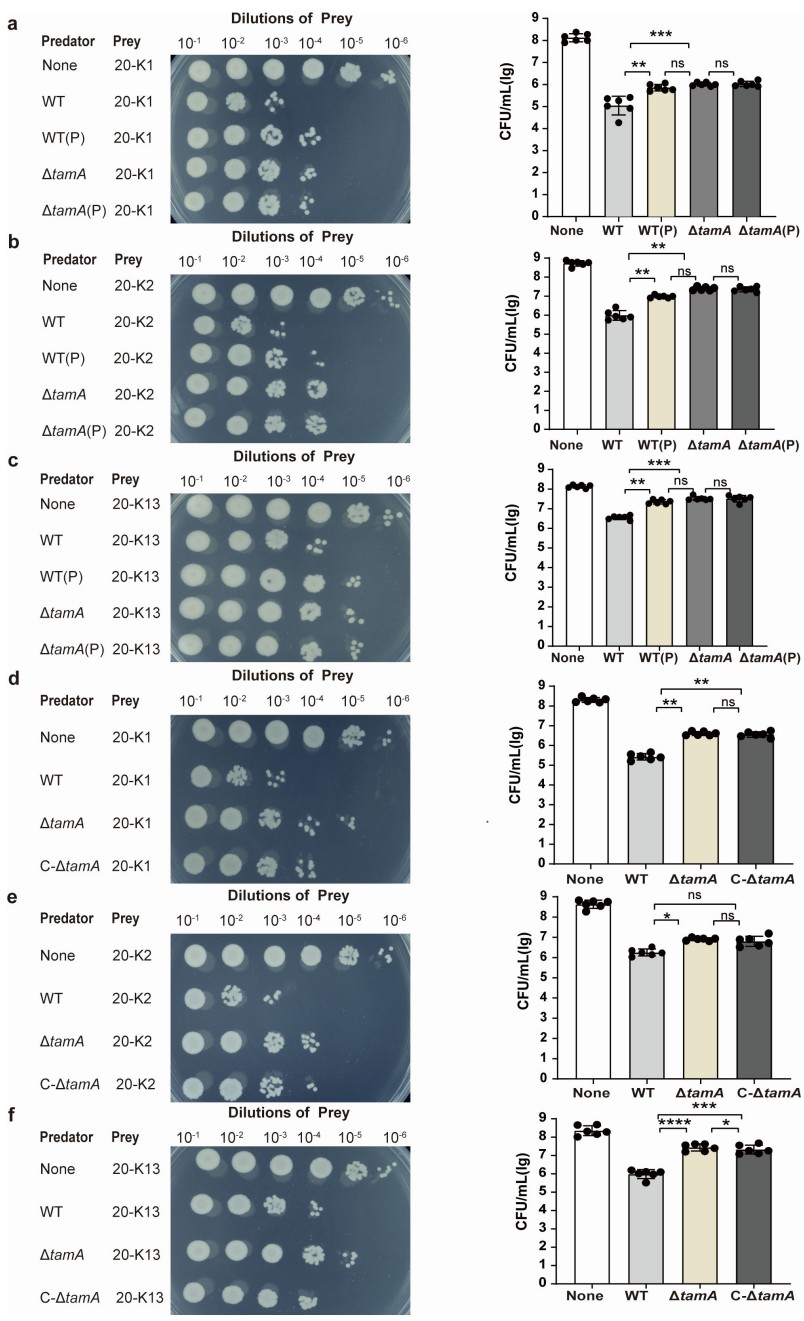

**FIG 5** PDX affects the intraspecies competition of *K. pneumoniae* by reducing the expression of TamA. The survival rate of the prey strain was measured after 20-K1 (a), 20-K2 (b), and 20-K13 (c) were incubated with WT, WT (P), Δ*tamA*, and Δ*tamA* (P) strains for 5 h. The survival of the prey strain was tested after 20-K1 (d), 20-K2 (e), and 20-K13 (f) were incubated with WT, Δ*tamA*, and C-Δ*tamA* strains for 5 h. Data are represented as means ± SD. Significance was assessed using a one-way ANOVA test followed by Tukey's *post hoc* test for multiple comparisons. *$P < 0.05$; **$P < 0.01$; ***$P < 0.001$; ****$P < 0.0001$; ns indicates not significant.

## PDX inhibits the adhesion of *K. pneumoniae* to Caco-2 and MLE cells by decreasing the expression of TamA

The adhesion ability of PDX-treated *K. pneumoniae* to Caco-2 and MLE cells was detected *in vitro*. The results showed that after co-culture with PDX and *K. pneumoniae* for 5 h, the adhesion ability to Caco-2 cells decreased by threefold (Fig. 6a and c), and the adhesion

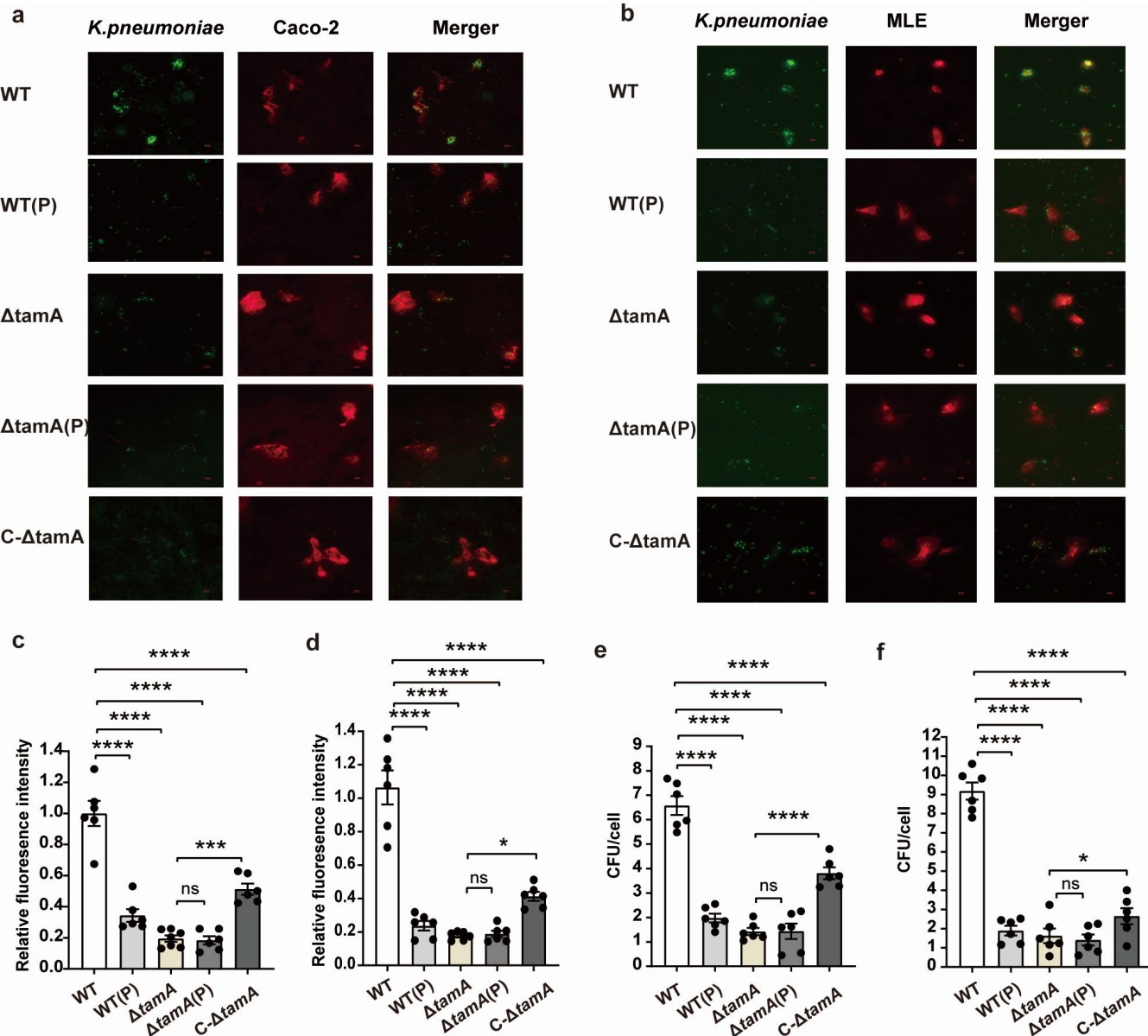

**FIG 6** Effect of PDX on adhesion of *K. pneumoniae* to Caco-2 and MLE cells. (a) The adhesion of *K. pneumoniae* to Caco-2 cells was observed using fluorescence microscope. (b) Observation of *K. pneumoniae* adhesion to MLE cells by fluorescence microscope. (c) Relative fluorescence intensity of adhesion Caco2 by *K. pneumoniae.* (d) Relative fluorescence intensity of adhesion MLE by *K. pneumoniae.* (e) Colony count of *K. pneumoniae* adhering to Caco-2 cells. (f) Colony counts of *K. pneumoniae* adherent MLE cells. Data are represented as means ± SEM. Significance was assessed using a one-way ANOVA test followed by Tukey's *post hoc* test for multiple comparisons. *$P < 0.05$; **$P < 0.01$; ***$P < 0.001$; ****$P < 0.0001$; ns indicates not significant.

ability to MLE cells decreased by fourfold (Fig. 6b and d). We further tested the adhesion of the Δ*tamA* and C-Δ*tamA* strains to the cells. The results show that compared with the WT strain, the adhesion ability of the Δ*tamA* strain to Caco-2 and MLE cells decreased significantly (Fig. 6e and f). However, the adhesion ability of the C-Δ*tamA* strain to cells has a recovery trend after the complementary *tamA* gene (Fig. 6e and f), which suggests that TamA is involved in the adhesion of *K. pneumoniae* to host cells. There is no significant difference in adhesion ability between Δ*tamA* and Δ*tamA*(P) strains to Caco-2 and MLE cells (Fig. 6e and f). These results illuminate that PDX intervention reduces the adhesion ability of *K. pneumoniae* to Caco-2 and MLE cells, which may be associated with downregulating the expression of TamA.

## Deletion of *tamA* attenuates the infection of *K. pneumoniae* in mice

To further confirm whether TamA is a main factor in inhibiting the infection of *K. pneumoniae* in mice, we constructed the infection mouse models with wild-type and Δ*tamA* strains by oral gavage and intranasal inoculation. The experimental design procedure is shown in Fig. 7a. The results reveal that after intranasal inoculation, *K. pneumoniae* load in the lungs of mice in the Δ*tamA*-KPN group is significantly reduced

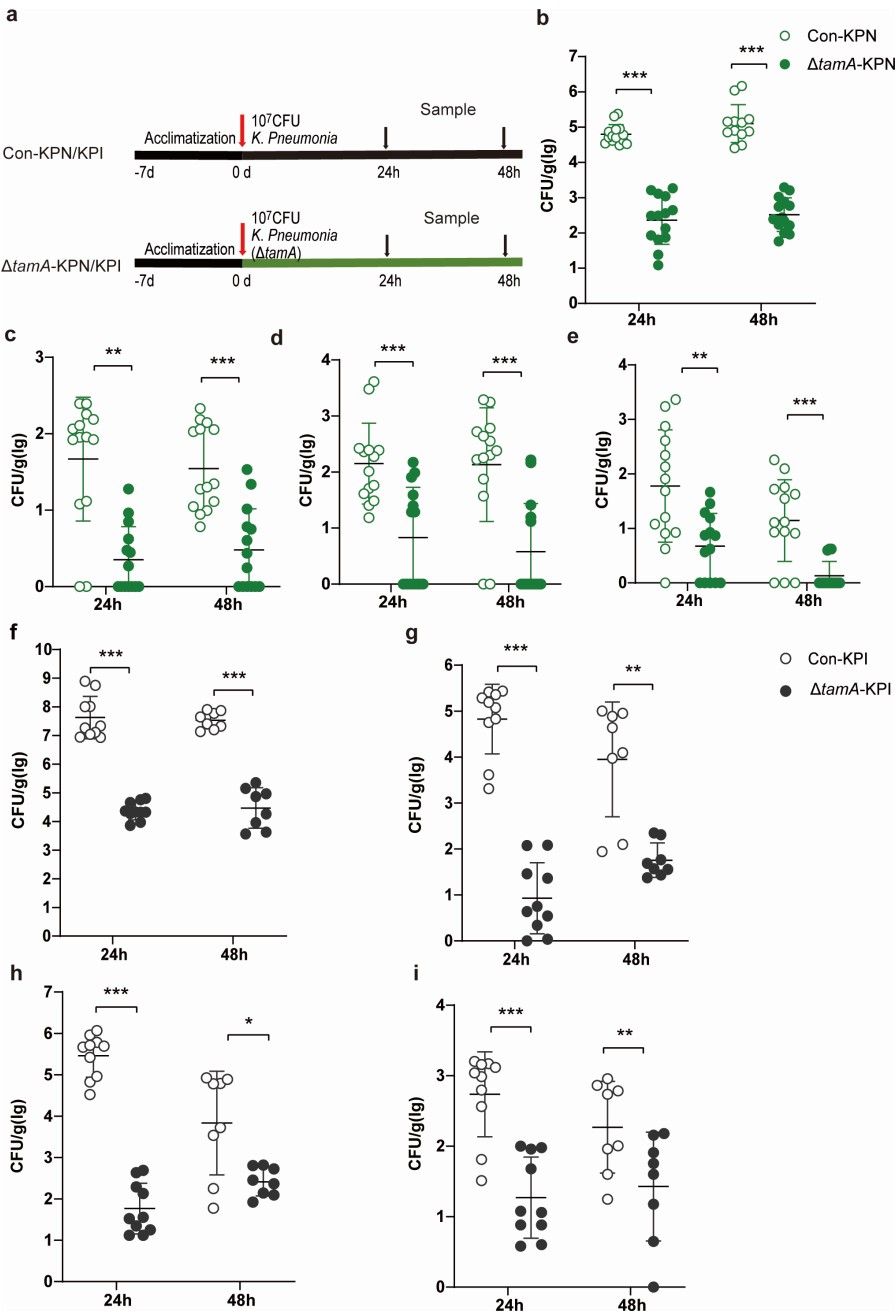

**FIG 7** Deletion of *tamA* reduces the infection of *K. pneumoniae* in mice. (a) The design of the experiment. After infecting mice with WT and Δ*tamA* strains by intranasal inoculation for 24 h and 48 h, respectively, the load of *K. pneumoniae* in the lungs (b), liver (c), spleen (d), and kidney (e) (*n* = 6–8). After infecting mice with WT and Δ*tamA* strains by oral gavage for 24 h and 48 h, respectively, the load of *K. pneumoniae* in intestine (f), liver (g), spleen (h), and kidney (i) (*n* = 6–8). Data are represented as means ± SEM. Significance was assessed using an unpaired *t*-test. *P < 0.05; **P < 0.01; ***P < 0.001; ****P < 0.0001; ns indicates not significant.

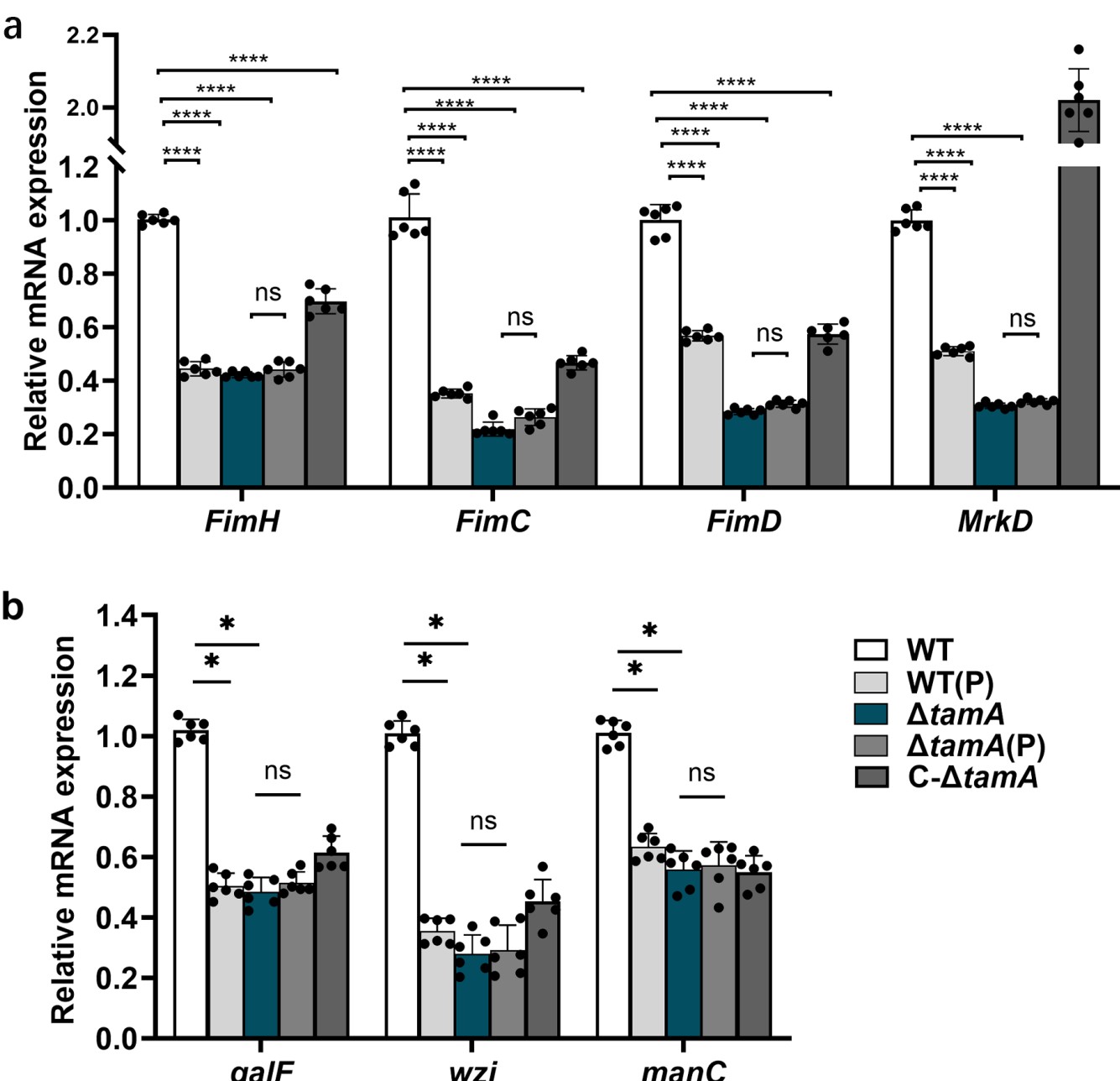

**FIG 8** PDX affects the expression of the capsule synthesis genes and adhesion-related factors of *K. pneumoniae* through TamA. (a) The mRNA expression levels of adhesion molecule gene (including *FimH*, *FimC*, *FimD*, and *MrkD*) in WT, WT (P), Δ*tamA*, Δ*tamA* (P), and C-Δ*tamA* strains. (b) The expression of the capsule synthesis genes, including *galF*, *wzi*, and *manC*, in WT, WT (P), Δ*tamA*, Δ*tamA* (P), and C-Δ*tamA* strains. Data are represented as means ± SD. Significance was assessed using a one-way ANOVA test and *t*-test, followed by Tukey's *post hoc* test for multiple comparisons. *$P < 0.05$; **$P < 0.01$; ***$P < 0.001$; ****$P < 0.0001$; ns indicates not significant.

compared with the Con-KPN group (Fig. 7b). *K. pneumoniae* load in the liver, spleen, and kidney is significantly decreased in the Δ*tamA*-KPN group vs the Con-KPN group (Fig. 7c through e). At 24 h and 48 h post-oral gavage with *K. pneumoniae* infection, the *K. pneumoniae* load in the intestine, liver, spleen, and kidney in the Δ*tamA*-KPI group is significantly lower than that of the control group (Fig. 7f through j). Our findings demonstrate that eradicating the *tamA* gene diminishes *K. pneumoniae* infection in mice.

## PDX affects the expression of the capsule synthesis genes and adhesion factors of *K. pneumoniae* by regulating TamA

Why does PDX downregulate the expression of TamA, and how is this associated with reducing the adhesion ability of *K. pneumoniae*? To address this problem, we investigated the gene expression levels of major adhesion molecules in *K. pneumoniae*. The results show that compared with the WT strain, the expressions of adhesion molecules, including *FimH*, *FimC*, *FimD*, and *MrkD*, are significantly downregulated after PDX treatment (Fig. 8a). The *fimH*, *fimC*, *fimD*, and *mrkD* genes of Δ*tamA* and Δ*tamA* (P) strains are also significantly downregulated, but there are no statistical differences between the two groups (Fig. 8a). However, compared with Δ*tamA*, the expression of these adhesion molecules has the restoring trend in the C-Δ*tamA* strain (Fig. 8a). Given the observations indicating that PDX treatment and Δ*tamA* strains have produced less uronic acid than the WT, it is likely that expression of the capsule locus is reduced in these circumstances. Thus, we investigate the expression level of the 3 cps gene clusters, including *galF*, *wzi*, and *manC*, by qRT-PCR in WT (Δ*tamA*) grown in LB medium with or without PDX. As shown in Fig. 8b, the levels of *galF*, *wzi*, and *manC* are significantly reduced in the PDX co-culture compared to the WT strains. The Δ*tamA* strain also shows a lower expression level of *galF*, *wzi*, and *manC* than that of the WT strains, endorsing the similar trends observed in the Δ*tamA* mutant with the PDX co-culture. A slight increase in the mRNA level of the *galF* and *wzi* genes was found in the C-Δ*tamA* strain, but there is no significant difference from the Δ*tamA* mutant. These results suggest that PDX reduces the expression of the capsule synthesis genes and adherence-related factors by downregulating the TamA levels.

## DISCUSSION

PDX is a special water-soluble dietary carbohydrate fiber that has low calories, stability, and high tolerance. It affects the absorption of host nutrients, composition of gut microorganisms, immune function, postprandial blood glucose, and blood lipids (28). PDX is known to be a carbon source that encourages the growth of beneficial bacteria like *Bifidobacterium* and *Lactobacillus*, simultaneously preventing harmful bacterial colonization in the intestine (26, 33). However, the emphasis of the current study is mainly focused on the indirect mechanism of PDX, which inhibits the infection of harmful bacteria. The exact mechanism of direct PDX resistance to the pathogens has not been reported yet. Therefore, keeping in view the direct effect of PDX on pathogens, the present study focuses on exploring how PDX reduces the infection of *K. pneumoniae* in mice.

In the current study, to assess the effect of PDX on the infection of *K. pneumoniae* in the host, mice were infected with WT and WT(P) strains by intranasal inoculation and oral gavage. *K. pneumoniae* load in the lungs, livers, spleens, kidneys, and intestines of the infected mice was explored at 24 h and 48 h after post-infection. The results showed that PDX significantly reduced the infection of *K. pneumoniae* in mice. *In vitro* study results show that the co-culture of PDX and *K. pneumoniae* did not affect the growth and morphological characteristics. Therefore, we hypothesize that PDX may affect the function of *K. pneumoniae* at the molecular level.

To explore the molecular mechanism of PDX's effect on the infection of *K. pneumoniae* in mice, RT-PCR was performed on total RNA extracted from WT and WT(P) strains to compare their gene expression profiles. It indicates that the expression of TamA of WT(P) was significantly lower than that of the WT strain. Further investigation at the protein level shows that PDX can reduce the expression of the membrane protein TamA. Could these results be due to other sugars (mono-, di-, or polysaccharides)? We selected four kinds of prebiotics (including resistant dextrin, PDX, galactooligosaccharides, and FOS) to investigate their effect on the *TamA* expression of *K. pneumoniae*. The results show that compared with the control group, resistant dextrin and galactooligosaccharides can improve the expression of *tamA*, while PDX and fructooligosaccharides can reduce the

expression of *tamA*, and PDX has a more significant effect on *tamA* expression compared with fructooligosaccharides (Fig. S1). Subsequently, PDX concentrations were screened to optimize the dose for its maximal effect on TamA. The results show that 5% PDX has a greater effect in reducing the TamA expression of *K. pneumoniae* (Fig. S2). Therefore, we selected a 5% concentration of PDX as the optimal intervention dose for this study.

To verify that the PDX effect on the adhesion of *K. pneumoniae* is achieved by reducing the expression of TamA, we constructed Δ*tamA* and C-Δ*tamA* strains and investigated the capsule production, biofilm formation ability, competition, and cell adhesion capacity. The results showed that the capsule production, the biofilm formation ability, the interspecific and intraspecific competitive ability, and the cell adhesion ability of *K. pneumoniae* co-culturing with PDX were decreased as compared to the wild-type strain, and Δ*tamA* also exhibited the same experimental results. Furthermore, when Δ*tamA* was pretreated with PDX, there was no additional reduction in these phenotypes. These results suggest that the knockout of *tamA* might have an impact on the survival of pathogenic bacteria, which aligns with previous reports (32). It is worth noting that when the complement plasmid carrying the *tamA* gene was introduced into the Δ*tamA* strain, the C-Δ*tamA* strain exhibited partial restoration of capsule production, biofilm formation, competitiveness, and cell adhesion. However, these functions did not fully return to the levels seen in the wild-type organism. This finding indicates that the plasmid-borne *tamA* gene was unable to completely restore the native *tamA* activity. We attribute this incomplete functional restoration to the complementary gene residing on an extrachromosomal plasmid rather than being integrated into the host genome, leading to a diminished *tamA* expression level in the C-Δ*tamA* strain compared to the wild type. Our data also confirm that PDX decreased the infection of *K. pneumoniae* in mice, which is directly related to TamA expression.

It is further validated *in vitro* experiment that PDX reduced the adhesion of *K. pneumoniae* to intestinal and pulmonary epithelial cells by downregulating TamA. To further evaluate the role of TamA in mediating the infection of *K. pneumoniae* in mice, we infected the mice with WT and Δ*tamA* strains, respectively. It was found that the deletion of *tamA* significantly reduced the infection of *K. pneumoniae* in mice compared with the WT strain, which is consistent with a previously reported study on the deletion of *tamA* affecting *Edwardsiella tarda* infection in the host (30). The present study explicitly demonstrates that PDX exhibits no inhibitory effects on the growth of both wild-type and ΔTamA *K. pneumoniae* strains. Instead, PDX reduces bacterial burden in the host by downregulating TamA expression. The precise molecular mechanism through which PDX interacts with TamA remains unresolved in the current experimental framework. We need additional experiments to further unveil this interaction. TamA serves as an essential subunit of the type II secretion system assembly (TAM), a class of outer membrane channel proteins. Structural or functional perturbations in TAM impair the efficient synthesis of outer membrane pili, thereby reducing bacterial adhesion and host colonization capacity. The finding aligns with prior studies demonstrating that TAM dysfunction disrupts pilus assembly and virulence factor secretion in gram-negative pathogens (37–39). The present data illustrate that the deletion of *tamA* inhibits the expression of the adhesion factors, including *FimH*, *FimC*, *FimD*, and *MrkD*. This finding aligns with previous literature showing that TAM mediates the assembly of *FimD* and other usher proteins in *E. coli* (40). It was therefore hypothesized that this effect may be influenced by the structure of TAM. Disruption or downregulation of TAM impairs outer membrane protein transport, leading to their periplasmic accumulation. This buildup then triggers negative feedback, which affects the expression of other outer membrane proteins.

The colonization of *K. pneumoniae in vivo* is a prerequisite for infection. Colonization of *K. pneumoniae* relies on its adhesion factor, primarily mediated by pili like type I and type III (41–43). Type I pili are mannose sensitive, which is encoded by the *Fim* (A, B, C, D, E, F, G, and H) gene cluster. *FimH*, located at the pili's tip, is an adhesin protein that can bind to mannose trisaccharides in host glycoproteins and mediate the bacterial adhesion

and invasion into the host cells (43). Type III pili are mannose insensitive and are encoded by the *Mrk* (A, B, C, D, E, and F) gene cluster. The *MrkD* subunit is also the apical adhesin protein of pili. Type III pili primarily facilitate bacterial adhesion to endothelial, respiratory epithelial, and urogenital epithelial cells (44). Reported literature illustrates that adhesion factors play a key role in the infection of *K. pneumoniae* in the host (45). Thus, when the expression of *FimH* and *MrkD* is downregulated, the formation ability of pathogenic biofilm will also be reduced (46–48). However, the current study reveals a link between the downregulation of adhesion factors (including *FimH*, *FimC*, *FimD*, and *MrkD*) and a reduced *K. pneumoniae* load mediated by TamA. Therefore, this study suggests that TamA may reduce the infection of *K. pneumoniae* in the host by affecting the expression of adhesion-related molecules.

Among the virulence factors identified in *K. pneumoniae*, the capsular polysaccharide (CPS) is considered the major determinant for *K. pneumoniae* virulence. A clear link between CPS and virulence has been demonstrated in multiple murine models of *K. pneumoniae* infection, including pneumonia and urinary tract infection (49). Traditionally, the hypermucoviscosity (hmv) has been closely associated with hvKp and is attributed to overproduction of CPS. Although early studies suggested that the hmv may not be solely due to overproduction of CPS, some research has shown that the absence of CPS can lead to a significant loss of *K. pneumoniae* hypermucoviscosity (50, 51). The bacterial capsule serves a variety of functions, including enhancing virulence, promoting adhesiveness, facilitating biofilm formation, and protecting against the immune response of the hosts. The CPS biosynthesis is not only regulated by genes but also influenced by environmental factors. Lin et al. reported that environmental glucose stimulates the production of capsular polysaccharide, a process regulated by cyclic adenosine monophosphate and recombinant chemokine C proteins. They suggest that *K. pneumoniae* may increase CPS production to enhance its persistence in the bloodstream of diabetic patients (52). However, we observed that PDX suppresses capsule production in *K. pneumoniae*, a phenomenon linked to TamA expression. The *cps* gene cluster of *K. pneumoniae* contains 19 open reading frames organized into three transcription units, namely, orf1–2, orf3–15, and orf16–17 (53). In the present study, we examined the expression of the 3 cps gene clusters, including *galF*, *wzi*, and *manC*. The results showed that PDX and TamA deletion could reduce the expression of genes. Prior research indicates that various regulators, like *RcsA/B*, *RmpA/A2*, *KvhA*, *KvgA*, and *Fur*, govern the transcription of CPS biosynthesis genes within *K. pneumoniae* (54–56). We also observed that TamA mediates the expression of cps genes in *K. pneumoniae*. The specific mechanism requires further study in the future. This study demonstrates that PDX reduces *K. pneumoniae* infection in mice, potentially by inhibiting TamA expression, which further reduces the synthesis and secretion of adhesion factors (including *FimH*, *FimC*, *FimD*, and *MrkD*) and the capsular polysaccharide. Thus, dietary PDX supplementation is a viable method for both reducing the *K. pneumoniae* burden and preventing infection.

## Conclusions

In conclusion, the present study reveals that (i) pretreatment of *K. pneumoniae* with PDX can significantly reduce the infection in the lung, liver, spleen, and intestinal tract of mice; (ii) pretreatment of *K. pneumoniae* with PDX does not alter the growth characteristics and morphological structure but can significantly reduce the expression of TamA; (iii) PDX inhibits the capsular production and the biofilm formation of *K. pneumoniae* and weakens the interspecific and intraspecific competitiveness with other Enterobacteriaceae, which ultimately decreases the adhesion ability to Caco-2 and MLE cells by downregulating the expression of TamA; and (iv) deletion of *tamA* significantly lessens the burden of *K. pneumoniae* in the lungs and intestines of mice. The potential mechanism is that deletion of the *tamA gene* decreases the expression of capsular and adhesion molecules, including *galF*, *wzi*, *manC*, *FimH*, *FimC*, *FimD*, and *MrkD*.

## ACKNOWLEDGMENTS

The authors wish to acknowledge funding from the Henan Province Science and Technology Research and Development Plan joint fund (industry) major project (grant 235101610004), the Science and Technology Research Project of Henan Province (grant 252300421398), the project of Henan Collaborative Innovation Center of Prevention and Treatment of Mental Disorders (grant XTkf10), and the Open Program of the Henan Key Laboratory of Biological Psychiatry (Program No ZDSYS2023004).

L.S. and H.Z.: investigation and writing. L.S., N.L., and Y.Y.: experiment and data analysis. H.M.I.: editing and review. D.L. and L.L.: software. C.W. and F.Y.: funding acquisition and project administration. All authors read and approved the final manuscript.

## AUTHOR AFFILIATIONS

[1]Henan Collaborative Innovation Center of Prevention and Treatment of Mental Disorders, Henan Key Laboratory of Biological Psychiatry, The Second Affiliated Hospital of Xinxiang Medical University, Xinxiang, China

[2]Department of Pathogenic Biology, School of Basic Medical Science, Xinxiang Medical University, Xinxiang, China

[3]Faculty of Veterinary and Animal Sciences, Muhammad Nawaz Shareef University of Agriculture, Multan, Pakistan

## AUTHOR ORCIDs

Chuansheng Wang  http://orcid.org/0009-0000-1002-2391
Fan Yang  http://orcid.org/0000-0001-9582-1394

## AUTHOR CONTRIBUTIONS

Lin Su, Investigation, Methodology, Writing – original draft | Huajie Zhao, Methodology, Resources | Hafiz Muhammad Ishaq, Writing – review and editing | Ningning Liu, Investigation, Methodology | Yalan Yang, Methodology | Duan Li, Resources, Software | Liang Liu, Software | Chuansheng Wang, Funding acquisition, Project administration | Fan Yang, Conceptualization, Funding acquisition, Writing – review and editing

## ETHICS APPROVAL

All experiments were conducted according to the Declaration of Helsinki and were approved by the Animal Care and Use Ethics Committee of Xinxiang Medical University (No. XYLL-20230094).

## ADDITIONAL FILES

The following material is available online.

### Supplemental Material

**Supplemental material (Spectrum01017-25-s0001.pdf).** Fig. S1 to S3.

### Open Peer Review

**PEER REVIEW HISTORY (review-history.pdf).** An accounting of the reviewer comments and feedback.

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
