## [Reviewer comments · Microbiology Spectrum]

Microbiology Spectrum

Polydextrose reduces the infection of *Klebsiella pneumoniae* in mice by downregulating the expression of TamA

Lin Su, Huajie Zhao, Hafiz Muhammad Ishaq, Ningning Liu, Yalan Yang, Duan Li, Liang Liu, Chuansheng Wang, and Fan Yang

Corresponding Author(s): Fan Yang, Xinxiang Medical University

Review Timeline:

Submission Date:	April 16, 2025
Editorial Decision:	July 29, 2025
Revision Received:	August 22, 2025
Editorial Decision:	September 5, 2025
Revision Received:	September 6, 2025
Accepted:	September 15, 2025

Editor: Sadjia Bekal

Reviewer(s): The reviewers have opted to remain anonymous.

Transaction Report:

DOI: <https://doi.org/10.1128/spectrum.01017-25>

Re: Spectrum01017-25 (**Polydextrose reduces the infection of *Klebsiella pneumoniae* in mice by downregulating the expression of TamA**)

Dear Prof. Fan Yang:

Thank you for the privilege of reviewing your work. Below you will find my comments, instructions from the Spectrum editorial office, and the reviewer comments.

Revision Guidelines

Sincerely,
Sadjia Bekal
Editor
Microbiology Spectrum

Polydextrose reduces the infection of *Klebsiella pneumoniae* in mice by downregulating the expression of TamA.

This is revised manuscript by Su et al that focuses on pretreatment of *K. pneumoniae* with polydextrose (PDX), and its impact on colonization and virulence. They also perform several in vitro experiments to look at the virulence traits of *K. pneumoniae* after treatment with PDX. While the authors have modified the manuscript there are concerns that require the authors attention.

1. While the authors say in their response that they tested several sugar molecules and their impact on TamA expression and also tested different concentrations of PDX. That data is neither present in the response as well as in the manuscript, where this information is critical to understand why they are trying to test PDX.
2. While they provide additional information about the strain used. They never test whether its capsular polysaccharide expression or amount is impacted by PDX treatment. This was raised in the first round of review as well. And is an important experiment that can link their adhesion and in vivo results.
3. In figure 2 growth curves and TamA expression is shown. Those two results cannot be correlated with each other. Saying that PDX treatment does not alter growth but impacts TamA expression is not correct. Growth curves were performed after treatment with PDX, and not during treatment with PDX. Whereas TamA expression is done after treatment with PDX. The authors should carry out growth assays with PDX added.
4. In figure 8 they test adhesion factors and show that they might be impacted with PDX treatment. Capsule levels directly impact *K. pneumoniae*'s ability to adhere to epithelial cells and biofilm formation. So, it would be pertinent to see whether expression and physiological levels of capsule are impacted by either PDX treatment or TamA deletion.

Minor issues.

1. While the writing is much improved there are still many places where sentence structure and tense are incorrect. As an example line 58-60.
2. Tam is an acronym please state it completely before using it. Also TamA is described as a molecule in line 243-244. Please remove that.
3. While some rationale is provided by the authors in response to previous comments. They still have not included that rationale for the killing and adhesion assays.
4. Conclusion in line 262-263 and 303-304 are suggesting linking PDX treatment and TamA levels to their phenotype. The Tam system has pleiotropic effects, and up to that point in their manuscript there is no direct linkage of PDX impacting only via TamA levels. Please tone down the conclusions.

Dear Editor,

We are thankful to the reviewers for their careful attention to our manuscript and for a number of valuable suggestions. We have addressed all the comments very carefully and hereby submit the revised version of the manuscript. The following is a point-by-point response to the reviewer's comments.

A point-by-point response to Reviewer :

Major comments:

1. While the authors say in their response that they tested several sugar molecules and their impact on TamA expression, and also tested different concentrations of PDX. That data is neither present in the response as well as in the manuscript, where this information is critical to understand why they are trying to test PDX.

Response: Firstly, in the pre-experimental phase, we selected 4 kinds of prebiotics (including resistant dextrin, polydextrose, galactooligosaccharides, and fructooligosaccharides) to evaluate their effect on the *TamA* expression in *K. pneumoniae*. The results show that, compared with the control group, resistant dextrin and galactooligosaccharides improve the expression of *TamA*, whereas polydextrose and fructooligosaccharides reduce the expression of *TamA*. However, polydextrose (PDX) has a more significant effect on *TamA* expression compared to fructooligosaccharides (Supplement Figure S1 in the manuscript). Then, we select different concentrations of PDX to screen for the optimal concentration that produces the strongest effect on *TamA*. The results demonstrate that 5% PDX has the strongest inhibitory effect on *TamA* expression in *K. pneumoniae* compared with other concentrations (Supplement Figure S2 in the manuscript). All this information has been incorporated in the discussion section of the manuscript.

2. While they provide additional information about the strain used. They never test

whether its capsular polysaccharide expression or amount is impacted by PDX treatment. This was raised in the first round of review as well. And is an important experiment that can link their adhesion and in vivo results.

Response: Thanks for the comment. According to the reviewer's instructions, we performed additional experiments to examine the effects of PDX or *TamA* deletion on the capsular polysaccharide production of *K. pneumoniae*. These results have been added to the results section of the manuscript.

3. In Figure 2 growth curves and TamA expression is shown. Those two results cannot be correlated with each other. Saying that PDX treatment does not alter growth but impacts TamA expression is not correct. Growth curves were performed after treatment with PDX, and not during treatment with PDX. Whereas TamA expression is done after treatment with PDX. The authors should carry out growth assays with PDX added.

Response: It seems that the reviewer may have misinterpreted the meaning conveyed in Figure 2 of the manuscript. When determining the growth curves, we set up 6 groups, namely the control group (only LB medium), the WT group (LB medium with *K. pneumoniae* inoculation), the WT(P) group (LB medium adding with PDX and *K. pneumoniae* inoculation), the $\Delta tamA$ group (LB medium with the $\Delta tamA$ mutant strain inoculation), the $\Delta tamA(P)$ group (LB medium adding with PDX and the $\Delta tamA$ mutant strain inoculation), and the C- $\Delta tamA$ group (LB medium with the *tamA* gene complementation strain inoculation). The blue color represents the growth curve of the wild strain after adding PDX in Figure 2a. Figure 2c illustrates the differences in TamA expression of *K. pneumoniae* after being cultured in media with or without PDX for a defined period. Thus, these two results should be able to be related to each other.

4. In figure 8 they test adhesion factors and show that they might be impacted with PDX

treatment. Capsule levels directly impact *K. pneumoniae*'s ability to adhere to epithelial cells and biofilm formation. So, it would be pertinent to see whether expression and physiological levels of capsule are impacted by either PDX treatment or TamA deletion.

Response: According to the reviewer's comments, we performed additional experiments to amplify the capsule synthesis genes, including *galF*, *wzi*, and *manC*, with RT-qPCR. It has been added in the result section as Figure 8b of the manuscript.

Minor issues.

1. While the writing is much improved there are still many places where sentence structure and tense are incorrect. As an example line 58-60.

Response: The entire manuscript has been edited by a native English speaker, and we carefully revised and corrected the tenses and sentence structure mistakes throughout the article.

2. Tam is an acronym please state it completely before using it. Also TamA is described as a molecule in line 243-244. Please remove that.

Response: We have corrected it in the manuscript.

3. While some rationale is provided by the authors in response to previous comments. They still have not included that rationale for the killing and adhesion assays.

Response: We have corrected the suggestion in the discussion section of the manuscript.

4. Conclusion in line 262-263 and 303-304 are suggesting linking PDX treatment and TamA levels to their phenotype. The Tam system has pleiotropic effects, and up to that point in their manuscript there is no direct linkage of PDX impacting only via TamA levels. Please tone down the conclusions.

Response: According to the reviewer's comments, we have corrected all the suggested

concerns in the whole manuscript.

In the end, we would like to thank the reviewers for their valuable suggestions regarding this article. We are submitting a copy of the revised manuscript indicating changes, so that the editor may check the revisions.

Sincerely,

Fan Yang

E-mail: yangf77@163.com

Re: Spectrum01017-25R1 (**Polydextrose reduces the infection of *Klebsiella pneumoniae* in mice by downregulating the expression of TamA**)

Dear Prof. Fan Yang:

Thank you for the privilege of reviewing your work. Below you will find my comments, instructions from the Spectrum editorial office, and the reviewer comments.

Revision Guidelines

Sincerely,
Sadjia Bekal
Editor
Microbiology Spectrum

Reviewer #1 (Comments for the Author):

The authors have substantially addressed the concerns raised by the reviewer. A minor point that they should address is the ability of the complement plasmid to restore the activity of TamA fully. This should be discussed in the discussion section.

A point-by-point response to Reviewer :

Reviewer #1 (Comments for the Author):

The authors have substantially addressed the concerns raised by the reviewer. A minor point that they should address is the ability of the complement plasmid to restore the activity of TamA fully. This should be discussed in the discussion section.

Response: According to the reviewer's comments, we have added the discussion about this issue in the manuscript.

Thanks again to the reviewers for their valuable suggestions regarding this article.

Sincerely,

Fan Yang

E-mail: yangf77@163.com

Re: Spectrum01017-25R2 (**Polydextrose reduces the infection of *Klebsiella pneumoniae* in mice by downregulating the expression of TamA**)

Dear Prof. Fan Yang:

Your manuscript has been accepted, and I am forwarding it to the ASM production staff for publication. Your paper will first be checked to make sure all elements meet the technical requirements. ASM staff will contact you if anything needs to be revised before copyediting and production can begin. Otherwise, you will be notified when your proofs are ready to be viewed.

Sincerely,
Sadjia Bekal
Editor
Microbiology Spectrum